# Expressive dynamics models with nonlinear injective readouts enable reliable recovery of latent features from neural activity

## Abstract

The advent of large-scale neural recordings has enabled new approaches that aim to discover the computational mechanisms of neural circuits by understanding the rules that govern how their state evolves over time. While these *neural dynamics* cannot be directly measured, they can typically be approximated by low-dimensional models in a latent space. How these models represent the mapping from latent space to neural space can affect the interpretability of the latent representation. Typical choices for this mapping (e.g., linear layer or MLP) lack the property of injectivity, meaning that changes in latent state may have no effect on neural activity. During training, non-injective readouts incentivize the invention of dynamics that misrepresent the underlying system and the computation it performs. Combining our injective Flow readout with prior work on interpretable latent dynamics models, we created the Ordinary Differential equations autoencoder with Injective Nonlinear readout (ODIN), which learns to capture latent dynamical systems that are nonlinearly embedded into observed neural firing rates via an approximately injective nonlinear mapping. We show that ODIN can recover nonlinearly embedded systems from simulated neural activity, even when the nature of the system and embedding are unknown. Additionally, we show that ODIN enables the unsupervised recovery of underlying dynamical features (e.g., fixed-points) and embedding geometry. When applied to biological neural recordings, ODIN can reconstruct neural activity with comparable accuracy to previous state-of-the-art methods while using substantially fewer latent dimensions. Overall, ODIN's accuracy in recovering ground-truth latent features and ability to accurately reconstruct neural activity with low dimensionality make it a promising method for distilling interpretable dynamics that can help explain neural computation.

## 1 Introduction

Recent evidence has shown that when artificial recurrent neural networks are trained to perform tasks, the rules that govern how the internal activity evolves over time (i.e., the network dynamics) can provide insight into how the network performs the underlying computation [1–4]. Given the conceptual similarities between artificial neural networks and biological neural circuits, it may be possible to apply these same dynamical analyses to brain activity to gain insight into how neural circuits perform complex sensory, cognitive, and motor processes [5–7]. However, unlike in artificial networks, we cannot easily interrogate the dynamics of biological neural circuits and must instead estimate them from observed neural activity.

Fortunately, advances in recording technology have dramatically increased the number of neurons that can be simultaneously recorded, providing ample data for novel population-level analyses of neural activity [8–10]. In these datasets, the activity of hundreds or thousands of neurons can often

Submitted to 37th Conference on Neural Information Processing Systems (NeurIPS 2023). Do not distribute.

be captured by relatively low-dimensional subspaces [11], orders-of-magnitude smaller than the total number of neurons. Neural activity in these latent spaces seems to evolve according to consistent sets of rules (i.e., latent dynamics) [12, 6]. Assuming no external inputs, these rules can be expressed mathematically as:

$$\mathbf{z}_{t+1} = \mathbf{z}_t + f(\mathbf{z}_t) \tag{1}$$

$$\mathbf{y}_t = \exp g(\mathbf{z}_t) \tag{2}$$

$$\mathbf{x}_t \sim \text{Poisson}(\mathbf{y}_t) \tag{3}$$

where $\mathbf{z}_t \in \mathbb{R}^D$ represents the latent state at time $t$, $f(\cdot) : \mathbb{R}^D \to \mathbb{R}^D$ is the vector field governing the dynamical system, $\mathbf{y}_t \in \mathbb{R}^N$ denotes the firing rates of the $N$ neurons, $g(\cdot) : \mathbb{R}^D \to \mathbb{R}^N$ maps latent activity into log-firing rates, and $\mathbf{x}_t \in \mathbb{R}^N$ denotes the observed spike counts at time $t$, assuming the spiking activity follows a Poisson distribution with time-varying rates given at each moment $t$ by $\mathbf{y}_t$.

Unfortunately, any latent system can be equivalently described by many combinations of dynamics $f$ and embeddings $g$, which makes the search for a unique latent system futile. However, versions of a latent system's dynamics $f$ and embedding $g$ that are less complex and use fewer latent dimensions can be much easier to interpret than alternative representations that are more complex and/or higher-dimensional. Models of latent dynamics that can discover simple and low-dimensional representations will make it easier to link latent dynamics to neural computation.

A popular approach to estimate neural dynamics [13–15] is to use neural population dynamics models (NPDMs), which model neural activity as a latent dynamical system embedded into neural activity. We refer to the components of an NPDM that learn the dynamics and embedding as the generator $\hat{f}$ and the readout $\hat{g}$, respectively. When modeling neural activity, the generator and readout are jointly trained to infer firing rates $\hat{\mathbf{y}}$ that maximize the likelihood of the observed neural activity $\mathbf{x}$.

Using NPDMs to estimate underlying dynamics and embedding implicitly assumes that good reconstruction performance (i.e., $\hat{x} \approx x$) implies interpretable estimates of the underlying system (i.e., $\hat{z} \approx z$, $\hat{f} \approx f$, $\hat{g} \approx g$). However, recent work has shown that when the state dimensionality of the generator $\hat{D}$ is larger than a system's latent dimensionality $D$, high reconstruction performance may actually correspond to estimates of the latent system that are overly complex or misleading and therefore harder to interpret [15]. Thus at present, reconstruction performance is seemingly an unreliable indicator for the interpretability of the learned dynamics.

This vulnerability to learning overly complex latent features might come from the fact that, in general, changes in the latent state are not obligated to have an effect on predicted neural activity. Thus, NPDMs can be rewarded for inventing latent activity that boosts reconstruction performance, even if that latent activity has no direct correspondence to the neural activity. A potential solution is to make the readout $\hat{g}$ injective, which obligates all latent activity to affect neural reconstruction. This would penalize any latent activity that is not reflected in the observed neural activity and puts pressure on the generator $\hat{f}$ and readout $\hat{g}$ to learn a more interpretable (i.e., simpler and lower dimensional) representation of the underlying system.

In addition, most previously used readouts $\hat{g}$ were not expressive enough to model diverse mappings from latent space to neural space, assuming the embedding $g$ to be a relatively simple (often linear) transformation (though there are exceptions [16–18]). Capturing nonlinear embeddings is important because neural activity often lives on a lower-dimensional manifold that is nonlinearly embedded into the higher-dimensional neural space [7]. Therefore, assumptions of linearity are likely to prevent NPDMs from capturing dynamics in their simplest and lowest-dimensional form, making them less interpretable than the latent features learned by NPDMs that can approximate these nonlinearities.

To address these challenges, we propose a novel architecture called the Ordinary Differential equation autoencoder with Injective Nonlinear readout (ODIN), which implements $\hat{f}$ using a Neural ODE (NODE [19]) and $\hat{g}$ using a network inspired by invertible ResNets [20–22, 19, 23]. ODIN approximates an injective nonlinear mapping between latent states and neural activity, obligating all latent state variance to appear in the predicted neural activity and penalizing the model for inventing excessively complex or high-dimensional dynamics. On synthetic data, ODIN learns representations of the latent system that are more interpretable, with simpler and lower-dimensional latent activity and dynamical features (e.g., fixed-points) than alternative readouts. ODIN's interpretability is also more robust to overestimates of latent dimensionality and can recover the nonlinear embedding of synthetic data that evolves on a simulated manifold. When applied to neural activity from a monkey performing

a reaching task with obstacles, ODIN reconstructs neural activity comparably to state-of-the-art recurrent neural network (RNN)-based models while requiring far fewer latent state dimensions. In summary, ODIN estimates interpretable latent features from synthetic data and can reconstruct biological neural recordings with high accuracy, making it a promising tool for understanding how the brain performs computation.

## 2 Related Work

Many previous models have attempted to understand neural activity through the lens of neural dynamics. Early efforts limited model complexity by constraining both $\hat{f}$ and $\hat{g}$ to be linear [24–26]. While these models were relatively straightforward to analyze, they often failed to adequately explain neural activity patterns [27].

Other approaches increased the expressiveness of the modeled dynamics $\hat{f}$. RNNs can learn to approximate complex nonlinear dynamics, and have been shown to substantially outperform linear dynamics models in reconstructing neural activity [27]. Unfortunately, RNNs implicitly couple the capacity of the model to the latent state dimensionality, meaning their ability to model complex dynamics relies on having a high-dimensional latent state. In contrast, NODEs can model arbitrarily complex dynamics of embedded dynamical systems at the dimensionality of the system [19, 15]. On synthetic data, NODEs have been shown to recover dynamics more accurately than RNN-based methods [28, 15]. In contrast to our approach, previous NODE-based models used a linear readout $\hat{g}$ that lacks injectivity. This can make the accuracy of estimated latent activity vulnerable to overestimates of the latent dimensionality (i.e., when $\hat{D} > D$) and/or fail to capture potential nonlinearities in the embedding $g$.

Early efforts to allow greater flexibility in $\hat{g}$ preserved linearity in $\hat{f}$, using feed-forward neural networks to nonlinearly embed linear dynamical systems in high-dimensional neural firing rates [16]. More recently, models have used Gaussian Processes to approximate nonlinear mappings from latent state to neural firing with tuning curves [17]. Other models have combined nonlinear dynamics models and nonlinear embeddings for applications in behavioral tracking [29] and neural reconstruction [18]. Additional approaches extend these methods to incorporate alternative noise models that may better reflect the underlying firing properties of neurons [16, 30]. While nonlinear, the readouts of these models lacked injectivity in their mapping from latent activity to neural activity.

Many alternative models seek to capture interpretable latent features of a system from observations. One popular approach uses a sparsity penalty on a high-dimensional basis set to derive a sparse symbolic estimate of the governing equations for the system [31]. However, it is unclear whether such sparse symbolic representation is necessarily a benefit when modeling dynamics in the brain. Another recent model uses contrastive loss and auxiliary behavioral variables to learn low-dimensional representations of latent activity [32]. This approach does not have an explicit dynamics model, however, so is not amenable to the dynamical analyses performed in this manuscript.

Normalizing flows – a type of invertible neural network – have recently become a staple for generative modeling and density estimation [20, 23]. Some latent variable models have used invertible networks to approximate the mapping from the latent space to neural activity [33] or for generative models of visual cortex activity [34]. To allow this mapping to change dimensionality between the latent space and neural activity, some of these models used a zero-padding procedure similar to the padding used in this manuscript (see Section 3.3.1), which makes the transformation injective rather than invertible [33, 23]. However, these previous approaches did not have explicit dynamics models, making our study, to our knowledge, the first to test whether injective readouts can improve the interpretability of neural population dynamics models.

## 3 Methods

### 3.1 Synthetic Neural Data

To determine whether different models can distill an interpretable latent system from observed population activity, we first used reference datasets that were generated using simple ground-truth dynamics $f$ and embedding $g$. Our synthetic test cases emulate the empirical properties of neural systems, specifically low-dimensional latent dynamics observed through noisy spiking activity [13, 35–37]. We sampled latent trajectories from the Arneodo system ($f$, $D = 3$) and nonlinearly embedded these trajectories into neural activity via an embedding $g$. We consider models that can recover the dynamics $f$ and embedding $g$ used to generate these data as providing an interpretable description of

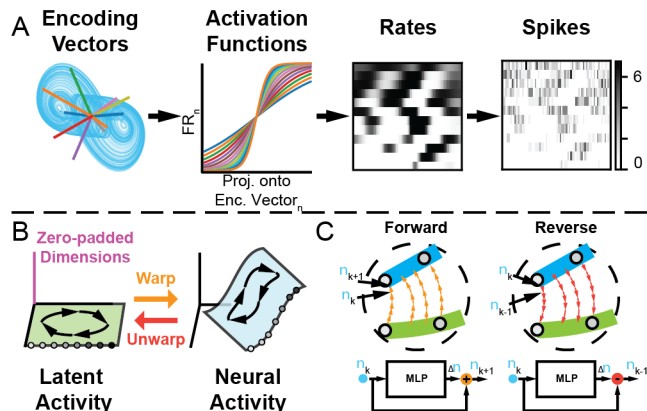

Figure 1: A) Synthetic neural data generation (left to right). Trajectories from the Arneodo system are projected onto random encoding vectors to compute activations at each timepoint. A scaled sigmoid nonlinearity is applied to convert the activations into firing rates. B) Zero-padded latent dynamics (green) are reversibly warped into higher-dimensional neural activity space (blue). C) The Flow readout maps from latent space to neural space by applying a sequence of $K$ small updates (parameterized by an MLP, bottom). Reverse pass maps from neural space to latent space and is implemented by serial subtraction of updates from the same MLP.

the latent system and its relation to the neural activity. Additional detail on data generation, models, and metrics can be found in the Supplementary Material.

We generated activations for $N$ neurons ($N = 12$) by projecting the simulated latent trajectories $\mathbf{Z}$ through a $3 \times N$ matrix whose columns were random encoding vectors with elements sampled from a uniform distribution $U[-0.5, 0.5]$ (Fig. 1A, left). We standardized these activations to have zero mean and unit variance and applied a different scaled sigmoid function to each neuron, yielding a matrix of non-negative time-varying firing rates $\mathbf{Y}$. The scaling of each sigmoid function was evenly spaced on a logarithmic scale between $10^{0.2}$ and $10$. This process created a diverse set of activation functions ranging from quasi-linear to nearly step-function-like behavior (Fig. 1A, Activation Functions).

We simulated spiking activity $\mathbf{X}$ by sampling from inhomogeneous Poisson processes with time-varying rate parameters equal to the firing rate $\mathbf{Y}$ of the simulated neurons (Fig. 1A, right). We randomly split 70-point segments of these trials into training and validation datasets (training and validation proportions were 0.8 and 0.2, respectively).

### 3.2 Biological Neural Data

We evaluated how well our model could reconstruct biological neural activity on a well-characterized dataset [38] included in the Neural Latents Benchmark (NLB) [27]. This dataset is composed of single-unit recordings from primary and pre-motor cortices of a monkey performing a visually-guided reaching task with obstacles, referred to as the Maze task. Trials were trimmed to the window [-250, 350] ms relative to movement onset, and spiking activity was binned at 20 ms. To compare the reconstruction performance of our model directly against the benchmark, we split the neural activity into held-in and held-out neurons, comprising 137 and 35 neurons, respectively, using the same sets of neurons as were used to assess models for the NLB leaderboard.

### 3.3 Model Architecture

We used three sequential autoencoder (SAE) variants in this study, with the main difference being the choice of readout module, $\hat{g}(\cdot)$. In brief, a sequence of binned spike counts $\mathbf{x}_{1:T}$ was passed through a bidirectional GRU encoder, whose final hidden states were converted to an initial condition $\hat{\mathbf{z}}_0$ via a mapping $\phi(\cdot)$. A modified NODE generator unrolled the initial condition into time-varying latent states $\hat{\mathbf{z}}_{1:T}$. These were subsequently mapped to inferred rates via the readout $\hat{g}(\cdot) \in \{\text{Linear}, \text{MLP}, \text{Flow}\}$. All models were trained for a fixed number of epochs to infer firing rates $\hat{\mathbf{y}}_{1:T}$ that minimize the negative Poisson log-likelihood of the observed spikes $\mathbf{x}_{1:T}$.

$$\mathbf{h}_T = \left[\mathbf{h}_{fwd} | \mathbf{h}_{bwd}\right] = \text{BiGRU}(\mathbf{x}_{1:T}) \tag{4}$$

$$\hat{\mathbf{z}}_0 = \phi(\mathbf{h}_T) \tag{5}$$

$$\hat{\mathbf{z}}_{t+1} = \hat{\mathbf{z}}_t + \alpha \cdot \text{MLP}(\hat{\mathbf{z}}_t) \tag{6}$$

$$\hat{\mathbf{y}}_t = \exp \hat{g}(\hat{\mathbf{z}}_t) \tag{7}$$

For models with Linear and MLP readouts, $\phi(\cdot)$ was a linear map to $\mathbb{R}^{\hat{D}}$. For models with Flow readouts, $\phi(\cdot)$ was a linear map to $\mathbb{R}^N$ followed by the reverse pass of the Flow (see Section 3.3.1). We unrolled the NODE using Euler's method with a fixed step size equal to the bin width and trained using standard backpropagation for efficiency. A scaling factor ($\alpha = 0.1$) was applied to the output of the NODE's MLP to stabilize the dynamics during early training. Readouts were implemented as either a single linear layer (Linear), an MLP with two 150-unit ReLU hidden layers (MLP), or a Flow readout (Flow) which contains an MLP with two 150-unit ReLU hidden layers. We refer to these three models as Linear-NODE, MLP-NODE, and ODIN, respectively.

### 3.3.1 Flow Readout

The Flow readout resembles a simplified invertible ResNet [23]. Flow learns a vector field that can reversibly transform data between latent and neural representations (Figure 1B). The Flow readout has three steps: first, we increase the dimensionality of the latent activity $\mathbf{z}_t$ to match that of the neural activity by padding the latent state with zeros. This corresponds to an initial estimate of the log-firing rates, $\log \hat{\mathbf{y}}_{t,0}$. Note that zero-padding makes our mapping injective rather than fully invertible (see [33, 23]). The Flow network then uses an MLP to iteratively refine $\log \hat{\mathbf{y}}_{t,k}$ over $K$ steps ($K = 20$) after which we apply an exponential to produce the final firing rate predictions, $\hat{\mathbf{y}}_t$. A scaling factor ($\beta = 0.1$) was applied to the output of the Flow's MLP to stabilize the dynamics during early training.

$$\log \hat{\mathbf{y}}_{t,0} = [\hat{\mathbf{z}}_t | \mathbf{0}]^T \tag{8}$$

$$\log \hat{\mathbf{y}}_{t,k+1} = \log \hat{\mathbf{y}}_{t,k} + \beta \cdot \text{MLP}(\log \hat{\mathbf{y}}_{t,k}) \tag{9}$$

$$\hat{g}(\hat{\mathbf{z}}_t) = \log \hat{\mathbf{y}}_{t,K} = \log \hat{\mathbf{y}}_t \tag{10}$$

We also use the approximate inverse of the Flow to transform the output of the encoders to initial conditions in the latent space via $\phi(\cdot)$. We approximate the inverse using a simplified version of the fixed-point iteration procedure described in [23]. Our method subtracts the output of the MLP from the state rather than adding it as in the forward mode (Fig 1C). From here, we trim the excess dimensions to recover $\hat{z} \in \mathbb{R}^{\hat{D}}$ (in effect, removing the zero-padding dimensions).

$$\log \hat{\mathbf{y}}_{t,k-1} = \log \hat{\mathbf{y}}_{t,k} - \beta \cdot \text{MLP}(\log \hat{\mathbf{y}}_{t,k}) \tag{11}$$

$$\hat{g}^{-1}(\log \hat{\mathbf{y}}_t) = [\log \hat{y}_{t,0,1}, \ldots, \log \hat{y}_{t,0,\hat{D}}]^T = \hat{\mathbf{z}}_t \tag{12}$$

The Flow mapping is only guaranteed to be injective if changes in the output of the MLP are sufficiently small relative to changes in the input (i.e., Lipschitz constants for the MLP that is strictly less than 1) [23]. The model can be made fully injective by either restricting the weights of the MLP (e.g., spectral norm [39]), or using a variable step-size ODE solver that can prevent crossing trajectories (e.g., continuous normalizing flows [19]. In practice, we found that using a moderate number of steps allows Flow to preserve approximate injectivity of the readout at all tested dimensionalities (Supp. Fig. 1).

### 3.4 Metrics and characterization of dynamics

All metrics were evaluated on validation data. Reconstruction performance for the synthetic data was assessed using two key metrics. The first, spike negative log-likelihood (Spike NLL), was defined as the Poisson NLL employed during model training. The second, Rate $R^2$, was the coefficient of determination between the inferred and true firing rates, averaged across neurons. We used Spike NLL to assess how well the inferred rates explain the spiking activity, while Rate $R^2$ reflects the model's ability to find the true firing rates. These metrics quantify how well the model captures the embedded system's dynamics (i.e., that $\hat{f}$ captures the system described by $f$), but give no indication of the interpretability of the learned latent representation (i.e., that the learned $\hat{f}$ is simple and low-dimensional).

To assess the interpretability of the latent activity inferred by the model $\hat{z}$, we used a previously published metric called the State $R^2$ [15]. State $R^2$ is defined as the coefficient of determination ($R^2$) of a linear regression from simulated latent trajectories $z$ to the inferred latent trajectories $\hat{z}$. State $R^2$ will be low if the inferred latent trajectories contain features that cannot be explained by an affine transformation of the true latent trajectories. We use this to assess the degree to which models can preserve the simplicity and low dimensionality of the embedded dynamics, thereby maintaining an interpretable latent representation. Together, high Rate $R^2$ and State $R^2$ indicate that the modeled latent activity reflects the simulated latent dynamics without inventing extra features that make the model harder to interpret (i.e., $\hat{z} \approx z$).

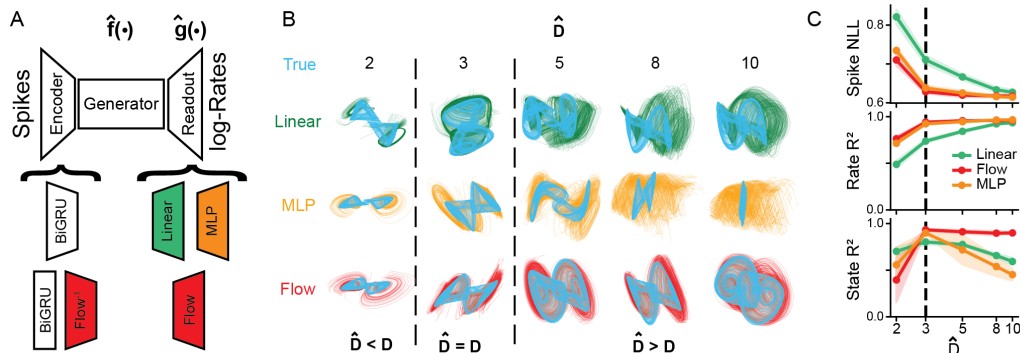

Figure 2: Flow-NODE (ODIN) recovers latent activity more accurately than alternative models and is robust to overestimates of latent dimensionality. A) Diagram of model readouts tested, including Linear (green), Flow (red), MLP (orange). B) Inferred latent activity of representative model at each state dimensionality $\hat{D}$. True latent activity (affine-transformed to overlay inferred latent activity) shown in light blue. C) All: Model metrics as a function of $\hat{D}$. Shaded areas represent one standard deviation around the mean. Dashed vertical line indicates $\hat{D} = 3$ Top: Spike NLL, Middle: Rate $R^2$, Bottom: State $R^2$.

As a direct comparison of the estimated dynamics $\hat{f}$ to the simulated dynamics $f$, we extracted the fixed-point (FP) structure from our trained models and compared it to the FP structure of the underlying system. We used previously published FP-finding techniques [40] to identify regions of the generator's dynamics where the magnitude of the vector field was close to zero, calling this set of locations the putative FPs. We linearized the dynamics around the FPs and computed the eigenvalues of the Jacobian of $\hat{f}$ to characterize each FP. Capturing FP location and character gives an indication of how closely the estimated dynamics resemble the simulated dynamics (i.e., $\hat{f} \approx f$).

To determine how well our embedding $\hat{g}$ captures the simulated embedding $g$, we projected the encoding vectors used to generate the synthetic neural activity from the ground-truth system into our model's latent space using the same affine transformation from ground-truth latent activity to inferred latent activity as was used to compute State $R^2$. We projected the inferred latent activity onto each neuron's affine-transformed encoding vector to find the predicted activation of each synthetic neuron. We then related the predicted firing rates of each neuron to its corresponding activations to derive an estimate of each neuron's activation function. Because the inferred latent activity is arbitrarily scaled/translated relative to the true latent activity, we fit an affine transformation from the predicted activation function to the ground-truth activation function. The coefficient of determination $R^2$ of this fit quantifies how well our models were able to recover the synthetic warping applied to each neuron (i.e., $\hat{g} \approx g$).

For the biological neural data, we measured model performance using two metrics from the Neural Latents Benchmark (NLB) [27], co-smoothing bits-per-spike (co-bps) and velocity decoding performance on predicted firing rates (Vel $R^2$). co-bps quantifies how well the model predicts the spiking of the held-out neurons, while Vel $R^2$ quantifies how well the denoised rates can predict the monkey's hand velocity during the reach. We compare these metrics to models from the NLB leaderboard. Of note, models submitted to NLB are assessed by their performance on a hidden test set, while our model performance is computed on the validation data.

## 4 Results

### 4.1 Finding interpretable latent activity across state dimensionalities with ODIN

We began by training Linear-, MLP-, and Flow-NODEs (i.e., ODIN) (Fig 2A) to reconstruct synthetic neural activity from the Arneodo system [41] and compared reconstruction performance (i.e. Spike NLL and Rate $R^2$) and latent recovery (i.e. State $R^2$) as functions of the dimensionality $\hat{D}$ of the state space. We trained 5 different random seeds for each of the 3 model types and 5 state dimensionalities (75 total models, model hyperparameters in Supp. Table 1). First, we observed that the Linear-NODE learned latent states that did not closely resemble the simulated latent activity, with all tested dimensionalities performing worse than either the Flow or the MLP readout at $\hat{D} = 3$ (Fig 2B,C, mean State $R^2 = 0.70$ for Linear vs. 0.89, 0.93 for MLP, Flow respectively). We also found that Linear-NODE required many more dimensions to reach the peak reconstruction performance

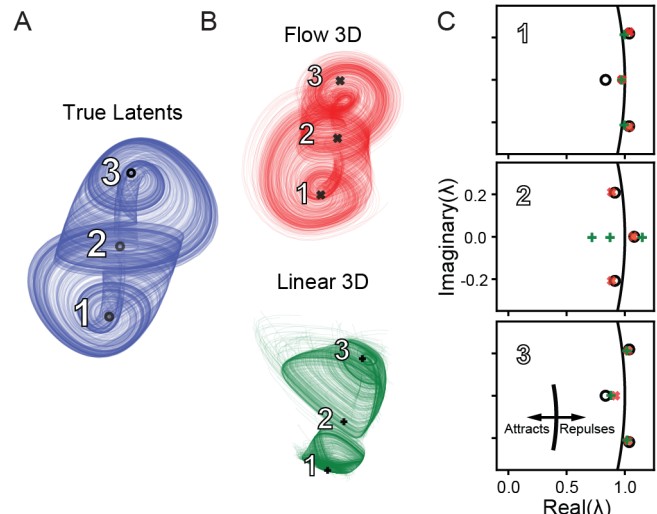

Figure 3: Flow-NODE (ODIN) recovers fixed-point properties accurately at the correct dimensionality. A,B) Representative latent activity and fixed-points from the true (blue, ○), ODIN (red, ×), and Linear (green, +) systems. Each fixed point is labeled with reference to C. C) Plots of the real vs. imaginary part of the eigenvalues of the Jacobian evaluated at each fixed point. Unit circle in the complex plane (black curve) shows boundary between attractive and repulsive behavior (the attractive and repulsive sides of the boundary are indicated by inset).

(Fig 2C, Rate $R^2$). These results demonstrate that models that are unable to account for nonlinear embeddings are vulnerable to learning more complex and higher dimensional dynamics than those learned by models with nonlinear readouts.

Next, we compared ODIN to MLP-NODE and found that at the correct dimensionality ($\hat{D} = 3$), these models had similar performance for both reconstruction and latent recovery. However, we found that as the dimensionality increased beyond the true dimensionality ($\hat{D} > 3$), the latent recovery of the MLP-NODE degraded rapidly while ODIN's latent recovery remained high (Fig 2C, as $\hat{D} > 3$). This result provides evidence that readouts that lack injectivity (like MLPs) tend to learn misleading latent activity that can make their representations less interpretable when the true dimensionality $\hat{D}$ is unknown.

### 4.2 Recovering fixed-point structure with ODIN

A common method to compare how well dynamics models capture the underlying dynamics from synthetic data is to examine the character and structure of the inferred fixed-points (FPs) to the FPs of the ground-truth system[15]. At a high-level, FPs enable a concise description of the dynamics in a small region of state-space around the FP, and can collectively provide a qualitative picture of the overall dynamical landscape. To obtain a set of candidate FPs, we searched the latent space for points at which the magnitude of the vector field $\|\hat{f}\|$ is minimized (as in [1, 40]). We computed the eigenvalues ($\lambda$s) of the Jacobian of $\hat{f}$ at each FP location. The real and imaginary components of these eigenvalues identify each FP as attractive, repulsive, etc.

We found that 3D ODIN models and 3D Linear-NODEs were both able to recover three fixed-points that generally matched the location of the three fixed points of the Arneodo system (Fig 3A), However, while ODIN was also able to capture the eigenspectra of all three FPs (Fig. 3B, red ×), the Linear-NODE failed to capture the rotational dynamics of the central FP (Fig 3B, middle column, green +). Both models were able to approximately recover the eigenspectra of outermost FPs of the system (Fig. 3B, left, right columns). We found that the MLP-NODE was also able to find FPs with similar accuracy to ODIN at 3D. These results show that the inability to model the nonlinear embedding can lead to impoverished estimates of the underlying dynamics $\hat{f}$.

### 4.3 Recovering simulated activation functions with ODIN

While obtaining interpretable dynamics is our primary goal,
models that allow unsupervised recovery of the embedding
geometry may provide additional insight about the compu-
tations performed by the neural system [42, 7]. For this
section, we considered a representative model from each
readout class with the correct number of latent dimensions
($D = 3$). We performed an affine transformation from the
ground truth encoding vectors into the modeled latent space
and computed the projection of the modeled latent activ-
ity onto the affine-transformed encoding vectors (Fig 4A).
From this projection, we derived an estimate of the activa-
tion function for each neuron, and compared this estimate
to the ground-truth activation function.

We found, as expected, that the linear readout was unable to
approximate the sigmoidal activation function of individual
neurons (Fig 4B, green). On the other hand, both ODIN
and MLP-NODE were able to capture activation functions
ranging from nearly linear to step function-like in nature
(Fig 4B, red, orange). Across all simulated neurons, we
found that ODIN more accurately estimated the activation
function of individual neurons compared to both Linear- and
MLP-NODEs (Fig 4C), suggesting that the injectivity of the
Flow readout allows more accurate estimation of nonlinear
embeddings.

### 4.4 Modeling motor cortical activity with ODIN

To validate ODIN's ability to fit neural activity from a bio-
logical neural circuit, we applied ODIN to the Maze dataset
from the Neural Latents Benchmark, composed of record-
ings from the motor and pre-motor cortices of a monkey
performing a reaching task (Fig. 5A). After performing hy-
perparameter sweeps across regularization parameters and
network size (Supp. Table 2), we trained a set of ODIN
and Linear-NODE models to reconstruct the neural activity
with a range of state dimensionalities $\hat{D}$. We visualized
the top 3 PCs of the condition-averaged latent trajectories
and predicted single-neuron firing rates for example models

Figure 4: Flow-NODE (ODIN) can re-
cover nonlinear activation functions of
neurons. A) True encoding vectors
(numbered lines over true latent ac-
tivity (blue)) were affine-transformed
into a representative model's latent
space. B) Inferred activation function
for two example neurons (columns),
color coded by readout type (Linear
= green, MLP = orange, Flow = red,
True = black). Plots show the predicted
firing rate vs. the activation of the se-
lected neuron. C) Comparison of the
$R^2$ values of the fits from B across
model types. Left: Flow vs. MLP.
Right: Flow vs. Linear

from each readout type. We found no visually obvious differences in the inferred latent trajectories
(Fig. 5B), but when we computed condition-averaged peri-stimulus time histograms (PSTHs) of
single neuron firing rates, we found that ODIN typically produced firing rate estimates that more
closely resembled the empirical PSTHs than those from the Linear-NODE (Fig. 5C).

Without access to a ground truth dynamics $f$ and embedding $g$ that generated these biological data, the
dimensionality required to reconstruct the neural activity was our primary measure of interpretability.
We computed co-bps –a measure of reconstruction performance on held-out neurons– for each model
and found that 10D ODIN models substantially outperformed Linear-NODE models, even when the
Linear-NODE had more than twice as many dimensions (10D ODIN: 0.333, vs 25D Linear: 0.287).
This suggests that ODIN's injective non-linear readout is effective at reducing the required latent
state dimensionality to capture the data relative to a simple linear readout.

We also compared ODIN to other models on the NLB leaderboard for this dataset [27, 43]. The best
reported AutoLFADS model (a RNN-based variational SAE with $\hat{D} = 100$) had only modestly higher
co-bps than the 10D ODIN (0.333 vs 0.355) [44]. These results suggest that ODIN is effective at
reducing the required dimensionality for neural reconstruction, which may provide more interpretable
latent representations than alternative models.

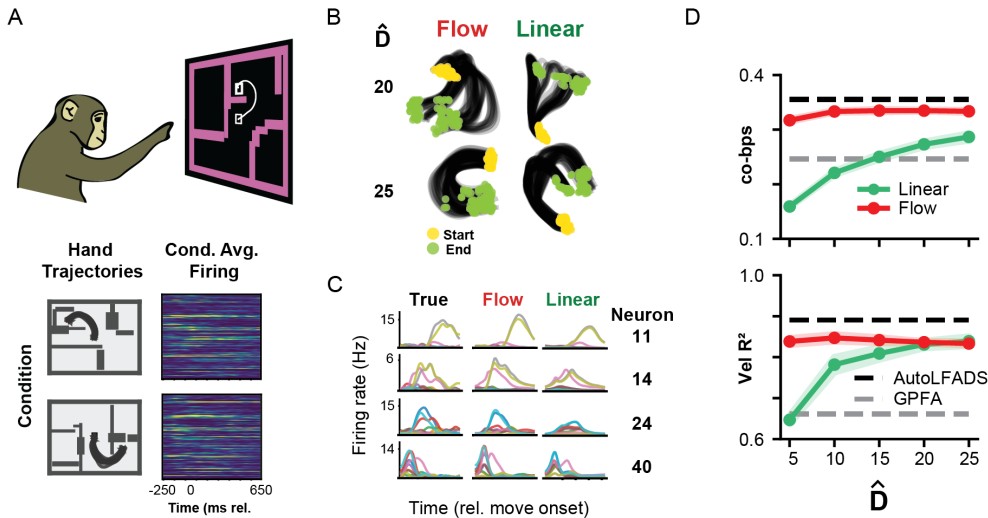

Figure 5: ODIN can reconstruct cortical activity with low-dimensional dynamics A) Top: Schematic of task [38] Bottom: example hand trajectories and condition-averaged firing rates aligned to move onset. B) Example condition-averaged latent activity from ODIN and Linear-NODE models applied to neural activity recorded during the Maze task. C) Example single-neuron peri-stimulus time histograms for ODIN and Linear-NODE models across conditions. D) Effects of latent state dimensionality $\hat{D}$ on reconstruction (top, co-bps) and decoding (bottom, Vel $R^2$) performance. Plot shows mean (point) and standard deviation (shading) of 5 randomly initialized models at each $\hat{D}$. Horizontal lines represent NLB performance by AutoLFADS (black) and GFPA (grey) [27].

## 5 Discussion

Dynamics models have had great success in reproducing neural activity patterns and relating brain activity to behavior [45, 27, 46]. However, it has been difficult to use these models to investigate neural computation directly. If neural population models could be trusted to find interpretable representations of latent dynamics, then recent techniques that can uncover computation in artificial networks could help to explain computations in the brain [1, 40, 47]. In this work, we created a new model called ODIN that can overcome major barriers to learning interpretable latent dynamical systems. By combining Neural ODE generators and approximately injective nonlinear readouts, ODIN offers significant advantages over the prior state-of-the-art, including lower latent dimensionality, simpler latent activity that is robust to the choice of latent dimensionality, and the ability to model arbitrary nonlinear activation functions.

Circuits in the brain are densely interconnected, and so a primary limitation of this work is that ODIN is not yet able to account for inputs to the system that may be coming from areas that are not directly modeled. Thus ODIN is currently only able to model the dynamics of a given population of neurons as an autonomous system. Inferring inputs is difficult due to ambiguity in the role of inputs compared to internal dynamics for driving the state of the system. While some RNN-based models have methods for input inference [45], more work is needed to develop solutions for NODE-based models. Injective readouts are an important step towards addressing the fundamental difficulties of input inference, as models without injective readouts can be incentivized to imagine latent features that are actually the result of inputs.

Interpretable dynamics derived from neural population recordings could answer critical scientific questions about the brain and help improve brain-machine interface technology. A potential negative consequence is that human neural interfaces combined with an understanding of neural computation might make it possible and profitable to develop strategies that are effective at influencing behavior. Future researchers should focus on applications of this research that are scientific and medical rather than commercial or political.

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
