# Expressive dynamics models with nonlinear injective readouts enable reliable recovery of latent features from neural activity
## *Supplementary Material*

# A  Datasets

## A.1  Simulated neural data

### A.1.1  Latent trajectories

We used the Arneodo system [1] to generate synthetic data because it exhibits mildly chaotic behavior (Lyapunov exponent equal to 0.243), it has a low-dimensional state space, and the regions around its fixed points are well-sampled by trajectories of the system. As demonstrated by [2], these properties allow recovery of latent dynamics in the absence of a nonlinear embedding. The Arneodo system is described by the following system of equations

$$\dot{x} = y \tag{1}$$
$$\dot{y} = z \tag{2}$$
$$\dot{z} = -ax - by - cz + dx^3 \tag{3}$$

where $a = -5.5$, $b = 4.5$, $c = 1.0$, and $d = -1.0$ [1].

The system was simulated using the `dysts` Python package, which offered well-reasoned standards for initial conditions, integration steps, and resampling frequency [3]. Initial conditions had been determined by running the model until the moments of the autocorrelation function were stationary. Integration steps had been chosen based on the highest significant frequency observed in the power spectrum. After integration, trajectories were resampled to contain 35 points per period, where period was based on the dominant frequency in the power spectrum.

### A.1.2  Embedding low-dimensional trajectories on a nonlinear manifold

We simulated neural activity by nonlinearly embedding the Arneodo trajectories as firing rates in the neural space. First, the trajectories were linearly projected into the neural space via a set of encoding vectors $\gamma_i$ and standardized for each neuron (see Methods). These activations $\mathbf{a}_i$ were passed through a sigmoid with input scaling $\eta_i$ and output scaling $b = 2$ to produce reasonable firing rates as follows:

$$\eta_{\mathbf{i}} = 10^{0.8 \cdot \frac{i-1}{N-1} + 0.2}, \tag{4}$$
$$\mathbf{y}_i = \psi_i(\mathbf{a}_i) = b \times \sigma(\eta_i \times \mathbf{a}_i), \quad i = 1, 2, \ldots, N. \tag{5}$$

where $\sigma(\cdot)$ denotes the sigmoid function. This resulted in a set of activation functions $\psi_i(\cdot)$ ranging from quasi-linear to step-like. The resulting rates $\mathbf{y}_i$ were used to parameterize a Poisson process, which was sampled to obtain spiking data for $N$ neurons ($N = 12$).

## A.2  Real neural data

The maze dataset was previously collected from the motor cortex of a monkey performing a reaching task [4]. This dataset has been widely used to characterize the dynamics of motor cortical activity [4–6]. In particular, these data are well-modeled by autonomous dynamics [6].

The monkey was trained to perform a delayed reaching task in which it had to maintain its hand at the center of a 2D maze displayed on a screen while a target was shown somewhere within the maze. After a randomly-timed delay, a go-cue was issued which prompted the monkey to move its hand from the center of the screen to the indicated target. Each trial also had a set of obstacles (i.e., the walls of a maze) with various configurations that required the monkey to produce reaches with varied trajectories, even when they were directed towards the same target. A total of 108 of these maze configurations (i.e., target and obstacle combinations) are included in this dataset.

Neural activity was recorded using two Utah arrays [7], one in the dorsal premotor (PMd) cortex and one in the primary motor cortex (M1) [4]. Threshold crossings were sorted offline. The dataset contained 182 neurons in total, of which 137 were included in the held-in set and the remaining 45 were part of the held-out set. The held-out neurons were used to calculate the co-smoothing bits-per-spike metric (F.2.1). The monkey's hand and cursor positions were recorded during the experiment (F.2.2).

These data were downloaded from the Distributed Archives for Neurophysiology Data Integration (DANDI, [8]). We binned spike counts at 20 ms and trialized and aligned the data to 250 ms before and 450 ms after movement onset. Further details can be found in [4, 5].

# B   Model training

## B.1   Simulated neural data

All weights were initialized from $\mathcal{U}(-\sqrt{k}, \sqrt{k})$, where $k = 1/\text{in\_features}$ for linear layers and $k = 1/\text{hidden\_size}$ for the GRU encoder weights. Dropout layers ($p = 0.05$) were inserted before and after the initial condition linear projection during training. We used the average Poisson negative log-likelihood (NLL) across neurons and time points as our training objective. Models were trained incrementally to improve the stability of training: rather than compute loss on the whole trajectory, we added groups of 5 new time steps every 75 epochs, up to the max of 70 steps. Models were trained by stochastic gradient descent using Adam for 3000 epochs. A single learning rate was shared for the optimizer of the encoder, generator, and readout weights for each model. Each generator was a NODE that contained an MLP with six hidden layers, each with 128 ReLU units. We did not find it necessary to regularize the weights for any of the models trained on the Arneodo system. We performed initial hyperparameter (HP) sweeps to determine HP ranges that resulted in good reconstruction performance as measured by Spike NLL (see Methods), and used the same HP set for models across state dimensionalities. HPs for models trained on the Arneodo system are given in Table S1.

Table S1: Training hyperparameters (Synthetic Data)

|  | Arneodo | | |
| --- | --- | --- | --- |
|  | Linear | MLP | ODIN |
| Batch Size | 650 | 650 | 650 |
| Learning Rate | 2e-3 | 1.88e-4 | 1.88e-4 |
| Encoder Hidden Size | 100 | 100 | 100 |
| Dropout | 0.05 | 0.05 | 0.05 |
| NODE Hidden Layers | 6 | 6 | 6 |
| NODE Hidden Size | 128 | 128 | 128 |
| Readout Hidden Layers | 0 | 2 | 2 |
| Readout Hidden Size | - | 150 | 150 |

## B.2  Real neural data

Weight initialization and dropout settings were identical to models trained on Arneodo. In addition to Poisson NLL, we also added regularization terms ($L_2$ norm on weights) and used different learning rates for the encoder, generator, and readout modules. We trained these models using Adam for 1500 epochs with the loss function given by Equation 6:

$$L(\mathbf{x}, \hat{\mathbf{y}}, \theta_E, \theta_G, \theta_R) = \text{PoissonNLL}(\mathbf{x}|\hat{\mathbf{y}}) + \lambda_E ||\theta_E||_2^2 + \lambda_G ||\theta_G||_2^2 + \lambda_R ||\theta_R||_2^2 \tag{6}$$

where $\mathbf{x}$ and $\hat{\mathbf{y}}$ represent the observed spiking activity and the predicted firing rates, respectively, and $\lambda_E, \lambda_G, \lambda_R$ represent the regularization coefficients for the $L_2$ regularization penalty applied to the model weights $\theta_E, \theta_G, \theta_R$ of the encoder, generator, and readout, respectively. To improve training stability, we also used different learning rates for each component of the model ($\alpha_E, \alpha_G, \alpha_R$). Specific parameters for models trained on the Maze dataset are given in Table S2.

Table S2: Training hyperparameters (Maze Data)

|  | Maze | |
| --- | --- | --- |
|  | Linear | ODIN |
| Batch Size | 64 | 64 |
| $\lambda_E$ | 1.6e-5 | 2.2e-6 |
| $\lambda_G$ | 1.6e-5 | 1.35e-9 |
| $\lambda_R$ | 1.6e-5 | 4.2e-6 |
| $\alpha_E$ | 5e-3 | 4e-4 |
| $\alpha_G$ | 5e-3 | 7e-4 |
| $\alpha_R$ | 5e-3 | 1.4e-4 |
| Encoder Hidden Size | 100 | 100 |
| Dropout | 0.05 | 0.05 |
| NODE Hidden Layers | 6 | 6 |
| NODE Hidden Size | 128 | 128 |
| Readout Hidden Layers | 0 | 3 |
| Readout Hidden Size | - | 128 |
| Number of Flow Steps | - | 25 |

# C  Approximate injectivity of Flow readout

To demonstrate the approximate injectivity of the Flow readout, we tested whether the readout could be inverted to recover the inferred latent activity. The readout mapping $\hat{g}$ should satisfy the following equations

$$\tilde{\mathbf{z}}_t = \hat{g}^{-1}(\hat{g}(\hat{\mathbf{z}}_t)) \tag{7}$$
$$\tilde{\mathbf{z}}_t \approx \hat{\mathbf{z}}_t \tag{8}$$

where $\hat{\mathbf{z}}_t$ is the inferred latent activity and $\tilde{\mathbf{z}}_t$ is the latent activity recovered by the reverse pass of the Flow.

We computed the $R^2$ between the inferred and recovered $\mathbf{z}_t$ for these models and found that our mappings were able to recover the inferred $\hat{\mathbf{z}}_t$ with average $R^2$ values across randomly initialized models of 0.997, 0.996, 0.990, and 0.988 at $\hat{D} = 3, 5, 8, 10$, respectively (Supplementary Figure S1).

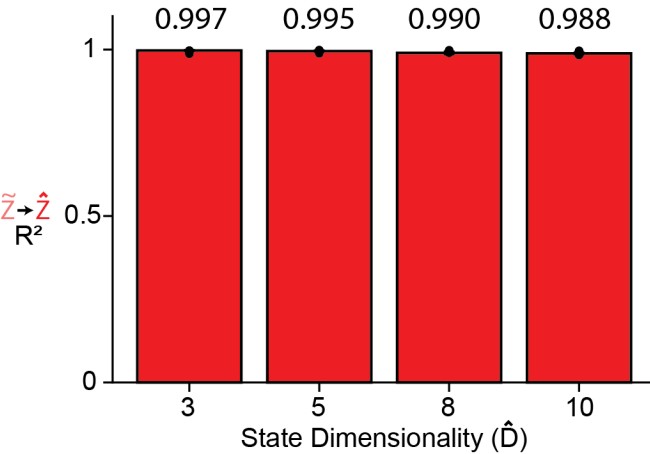

Figure S1: Injectivity of the Flow readout across state dimensionalities. Each bar indicates the mean value of 5 randomly initialized ODIN models for each state dimensionality. Results from individual models are plotted as points.

# D Fixed point finding and characterization

For each model (Linear-NODE, MLP-NODE and ODIN), we located fixed points (FPs) by finding the positions in the latent space that minimized the norm of the vector field via the objective $q = \frac{1}{2}\|\hat{f}\|_2^2$ [9, 10]. We initialized our search with 1024 randomly sampled initial states from along inferred latent trajectories. We used Adam with a learning rate of 5e-2 to minimize the $q$-value for each point independently over 10,000 iterations. Candidate points that did not achieve a $q$-value less than a magnitude of 7e-3 were excluded. As more than one candidate can approach the same FP, we combined candidate points that were within a specified distance, $\epsilon = 1$, from one another. In practice, points that were excluded had much larger $q$-values than the putative fixed points. We then linearized the dynamics around each FP and computed the system Jacobian to determine the stability and rotational character of the system around these FPs.

# E Compute resources

We used an internal computing cluster with a total of 30 Nvidia GeForce RTX 2080 Ti GPUs for model training. Each model trained on simulated neural data took approximately 3 hours to train, while each model trained on real biological data took approximately 1.5 hours to train. With 2 models training on each GPU, the 75 models included in Figs. 2, 3, and 4 took approximately 112.5 GPU-hours and the 50 models included in Fig. 5 took approximately 37.5 GPU-hours. FP finding was fast, requiring 1 minute for each model.

# F  Metrics

## F.1  Synthetic data metrics

### F.1.1  Rate reconstruction (Rate $R^2$)

We computed the coefficient of determination between true ($\mathbf{Y}$) and predicted ($\hat{\mathbf{Y}}$) rates for each neuron, and reported the average value across neurons.

$$\text{Rate } R^2 = R^2(\mathbf{Y}, \hat{\mathbf{Y}}) = \frac{1}{N} \sum_{i=0}^{N} 1 - \frac{\sum (\mathbf{y}_i - \hat{\mathbf{y}}_i)^2}{\sum (\mathbf{y}_i - \bar{\mathbf{y}}_i)^2}$$

### F.1.2  Latent state reconstruction (State $R^2$)

To compute State $R^2$, we concatenated a vector of ones with the true latent states ($\mathbf{Z_1}$), then used the pseudoinverse to find the optimal affine transformation from the true latents to the inferred latents ($\hat{\mathbf{Z}}$) (i.e., optimal linear estimation). We computed the coefficient of determination ($R^2$) between the true and inferred latent activity with the same equation as in F.1.1.

$$\mathbf{W}_z = \mathbf{Z}_1^\dagger \hat{\mathbf{Z}} \tag{9}$$

$$\text{State } R^2 = R^2(\hat{\mathbf{Z}}, \mathbf{Z}_1 \mathbf{W}_z) \tag{10}$$

### F.1.3  Activation function comparison

We developed a method for deriving an estimate of the inferred activation functions $\hat{\psi}_{\mathbf{i}}(\cdot)$ for a comparison to the true activation functions $\psi_{\mathbf{i}}(\cdot)$ (see Equation 5). We projected the true encoding vectors $\boldsymbol{\gamma}_i$ into the latent space of the model via the affine transformation $\mathbf{W}_z$ (see section F.1.2). We then used these encoding vectors $\hat{\boldsymbol{\gamma}}_i \in \mathbb{R}^{\hat{D}}$ to convert inferred latent states $\hat{\mathbf{Z}} \in \mathbb{R}^{T \times \hat{D}}$ into an activation $\hat{\mathbf{a}}_i \in \mathbb{R}^T$ for each neuron.

$$\hat{\boldsymbol{\gamma}}_i = \boldsymbol{\gamma}_{1,i} \mathbf{W}_z, \text{ for } i = 1, 2, \cdots, N \tag{11}$$

$$\hat{\mathbf{a}}_i = \hat{\mathbf{Z}} \cdot \hat{\boldsymbol{\gamma}}_i \tag{12}$$

To estimate the activation function for a given neuron $i$, we need pairs of inferred activations $\hat{a}_i$ and firing rates $\hat{y}_i$. For each neuron, we split firing rates into 20 quantiles and computed the corresponding median activation $\hat{a}_{i,1:20}^{med}$ and firing rate $\hat{y}_{i,1:20}^{med}$ within each quantile.

$$\hat{y}_{i,1:20}^{med}, \hat{a}_{i,1:20}^{med} = \text{Quantize}(\hat{\mathbf{y}}_i, \hat{\mathbf{a}}_i, 20) \tag{13}$$

We represented the inferred activation function $\hat{\psi}_i(\cdot)$ using these activation-firing rate pairs. We then performed the same procedure on the true rates and activations to find a similar representation of the true activation function $\psi_i(\cdot)$ for each neuron. To compare the true activation function $\psi(\cdot)$ to the estimated activation function $\hat{\psi}(\cdot)$, we combined the activations of each neuron $i$ and its corresponding firing rate as the columns of the matrices:

$$\hat{\boldsymbol{\Psi}}_i = \begin{pmatrix} \hat{\mathbf{a}}_i^{med} & \hat{\mathbf{y}}_i^{med} \end{pmatrix}, \quad \boldsymbol{\Psi}_i = \begin{pmatrix} \mathbf{a}_i^{med} & \mathbf{y}_i^{med} \end{pmatrix}$$

Because the inferred latent activity can be scaled and translated arbitrarily with respect to the true latent activity, we found the optimal affine transformation between $\hat{\boldsymbol{\Psi}}_i$ and $\boldsymbol{\Psi}_i$. We used the $R^2$ of this mapping to quantify the correspondence between the two activation functions $\hat{\psi}_i(\cdot)$ and $\psi_i(\cdot)$ for each neuron.

## F.2 Neural Latents Benchmark metrics

### F.2.1 Co-smoothing bits-per-spike (co-bps)

Large SAEs often have sufficient computational capacity to simply pass the spiking activity through the model (i.e., to learn the identity function) [11]. This can lead to models that obtain high reconstruction performance by only learning a trivial transformation, which would hinder learning of interpretable low-dimensional representations. A previously developed metric called co-smoothing bits-per-spike is sensitive to this form of overfitting because reconstruction performance is evaluated on a set of held-out neurons that are not visible to the encoders [5]. At a high-level, this metric quantifies how well the firing rates of the held-out neurons can be predicted from the spiking of the held-in neurons (see A.2). This metric is defined by Equation 14 for each held-out neuron.

$$\text{co-bps} = \frac{1}{n_s \log 2}(\mathcal{L}(\hat{\mathbf{y}}_{n,t}; \mathbf{x}_{n,t}) - \mathcal{L}(\bar{\mathbf{y}}_{n,:}; \mathbf{x}_{n,t})) \tag{14}$$

where $\bar{\mathbf{y}}_{n,:}$ is the mean firing rate for neuron $n$ across time, $n_s$ is the total number of spikes for that neuron, $\hat{\mathbf{y}}_{n,t}$ is the predicted firing rate from the model at time $t$, $\mathbf{x}_{n,t}$ represents the observed spiking of that neuron at time $t$, and $\mathcal{L}$ represents the Poisson log-likelihood. More information can be found in [5].

### F.2.2 Velocity decoding $R^2$

A common metric of performance is how well inferred firing rates can be used to predict behavioral variables, as this can be used downstream in decoding intent for clinical applications like brain-computer interfaces [12]. For the Maze dataset, hand velocity has been shown to be highly correlated with the neural firing in motor cortices. We compute this metric using the method from [5], in which a ridge regression model is trained to predict the observed hand velocity from inferred firing rates. The coefficient of determination ($R^2$) was then evaluated on validation data that was not used to train the ridge regression velocity decoder.

# G  Open-source packages used

- `torch` [13] (BSD license): Deep learning framework providing layer definitions, GPU acceleration, automatic differentiation, optimization, and more.

- `pytorch_lightning` (Apache 2.0 license): Lightweight wrappers for model training.

- `ray.tune` [14] (Apache 2.0 license): Distributed hyperparameter tuning.

- `dysts` [3] (Apache 2.0 license): Implementations for modeled dynamical systems.

- `fixed_point_finder` [10] (Apache 2.0 license): Inspiration for `torch`-based fixed point finder.

- `scikit-learn`[15] (BSD License): Implementations of linear regression models and principal component analysis.