# OpenReview forum: "Expressive dynamics models with nonlinear injective readouts enable reliable recovery of latent features from neural activity"
_NeurIPS.cc/2023/Conference — Submitted to NeurIPS 2023_

### Official Review · Reviewer_dR2R · 2023-06-12

**Soundness:** 3 good
**Presentation:** 3 good
**Contribution:** 3 good
**Rating:** 5
**Confidence:** 3

**Summary:**

This paper develops a new latent variable model for neural recordings ('ODIN') that uses a flow-based readout to map from the latent space to neural activity. The authors motivate their method by the injectivity of such flow-based methods and compare it to baseline models with linear and MLP readouts. ODIN exhibits superior performance on synthetic and biological datasets across a range of performance metrics.

**Strengths:**

It is important to improve the ability of our neural data analysis tools to capture the neural dynamics giving rise to our high-dimensional recordings. This paper uses new ideas from machine learning to this effect and develops a method that shows good performance across a synthetic and a biological dataset.

The authors are also very thorough in their characterization of the different models, where they include an impressive breadth of evaluations including reconstruction of neural activity, latent trajectories, activation functions, and learned fixed point structure.

**Weaknesses:**

As a reader/reviewer, I am left unconvinced that injectivity is a key problem with existing methods.

Linear models $Y = WX$ are for example injective provided that $W \in \mathbb{R}^{N \times D}$ is at least rank D, which will happen with high probability unless there are specific mechanisms to prevent it.

Similarly, while the paper does demonstrate that 'ODIN' performs better than MLP-NODE, it is not clear that this is due to limitations related to the injectivity of the MLP-based model. Indeed MLPs are perfectly capable of learning injective functions - and even if the mapping learned in this case is not injective, it is hard to know whether the performance difference is due to a lack of injectivity without exploring a much broader set of models and showing that injective models consistently outperform non-injective models (for example, the linear model considered in this work is injective with high probability, yet performs poorly).

Additionally, ODIN is motivated as being more interpretable than existing methods, but it uses a fairly complicated generative model, which makes the final learned mapping more opaque to the user. In this sense, it could be seen as less interpretable than simple linear methods, where the learned mapping is easily accessible and understandable (although this of course comes at the cost of a significant decrease in performance). Indeed, the authors state in L46 that "versions of a latent system's dynamics $f$ and embedding $g$ that are less complex and use fewer latent dimensions can be much easier to interpret than alternative representations that are more complex and/or higher dimensional." They do address the question of dimensionality, showing that ODIN captures neural data with fewer latent dimensions than linear methods, but this comes at a substantial increase in the complexity of $f$ and $g$.

**Questions:**

Is it possible to show that the superior performance of ODIN over MLP-NODE is due to ODIN learning an injective map?

What is the performance of MLP-NODE on the monkey Maze Dataset?

It is stated in several places that linear methods are not injective, but to the best of my knowledge this is only true in the special case where the readout matrix is not full rank.

**Limitations:**

The authors have addressed the limitations and potential negative impact of the work.

---

> ### Author Rebuttal · Authors · 2023-08-10
>
> We thank the reviewer for their detailed and insightful comments. While they were impressed by the breadth of our evaluation metrics, they were unconvinced by some of our arguments about the importance of readout injectivity for accurate latent inference.
>
> - I am left unconvinced that injectivity is a key problem with existing methods.
>
> Previous work [Sedler et al. 2023] established that both RNN- and NODE-based models consistently learn to invent superfluous latent features that help achieve better data reconstruction performance. This reduces their correspondence to the ground truth latent activity and thus hinders interpretability. By making our readout injective, we penalize unconstrained use of latent dimensions by requiring latent state variance to have a direct effect in the neural data space. To further demonstrate the benefits of injectivity, we show below that an alternative injective readout architecture (Invertible Neural Networks) shows similar advantages to the Flow over MLP readouts, giving compelling, if indirect, evidence that readout injectivity is the key innovation that improves the accuracy of latent recovery from synthetic neural recordings.
>
> - It is stated in several places that linear methods are not injective, but this is only true in the special case where the readout matrix is not full rank.
>
> The primary failing of Linear-NODE on these data is not a lack of injectivity, but poor reconstruction performance (see Fig R4) that arises because the linear readout cannot learn the true nonlinear embedding $g$. Previous research has shown that when $\hat{D}>D$, Linear-NODEs fit to linearly embedded data can improve reconstruction performance at the expense of latent recovery [1]. While this finding suggests that Linear readouts may indeed be non-injective when $\hat{D} >D$, that hypothesis was not explicitly tested in prior work.
>
> In Fig. R6, we demonstrate that the trade-off between reconstruction and latent recovery is accompanied by non-injective transformations in the linear readout. We trained Linear-NODE models with varying state dimensionality on a linearly-embedded Lorenz dataset and quantified their “effective rank” [Roy et al 2007], a continuous analog to the rank of a matrix that is robust to noise.
>
> The readouts’ effective rank saturated at less than 4D, suggesting that only around 4 (of 10 for $\hat{D}=10$) dimensions of the inferred latent activity significantly affect the predicted firing rates. When Linear-NODEs have excess state-dimensionality $\hat{D}$, those extra dimensions form an effective null-space which allows models to trade latent accuracy for reconstruction performance. We propose the addition of a figure to the main text of the revised manuscript that shows this effective rank quantification, and that Linear-NODE reconstruction improvements co-occur with both State $R^2$ deficits and the formation of ‘effective null-spaces’.
>
> - Is it possible to show that the superior performance of ODIN over MLP-NODE is due to ODIN learning an injective map?
>
> We agree that this is an important question whose answer could strengthen the impact of the work. First, we note that the combination of non-linearity and injectivity in the readout is the key contribution to this work, as the Linear-NODEs reconstruction performance was poor. To confirm that injectivity was the critical addition to non-linear readouts that made latent recovery more robust, we have found comparable results using an alternative injective architecture — an invertible neural network (INN) [Dihn et al. 2016]. We found that using the INN in place of the Flow with ODIN had comparable Rate $R^2$ and State $R^2$ to ODIN, and that State $R^2$ was stable as $\hat{D}$ increased beyond $D$ (see Fig. R7). This result further supports our claims that injective networks empirically promote robust latent recovery. We propose to include these figures in the supplementary material of the final manuscript.
>
> - Additionally, ODIN is motivated as being more interpretable than existing methods, but it uses a fairly complicated generative model...
>
> Our goal is not necessarily simple and interpretable architectures, but simple and interpretable model features. Since we are interested in understanding the role of dynamics in neural systems, our idea of interpretability is a low-dimensional latent dynamical system $\hat{f}$ that explains the temporal patterns in the data while preserving its underlying topology. While linear readouts may define a more straightforward relationship between the latent space and the neural data space, their simplicity/linearity requires $\hat{f}$ to capture any the nonlinearity in the data manifold, thereby hindering its interpretability (not to mention the drop in performance). To alleviate this confound, it is necessary to use nonlinear readouts to capture any such nonlinearity in the data manifold. Choosing a nonlinear readout with appropriate characteristics then becomes the challenge. In this work we have demonstrated the robustness that the Flow provides over the more standard MLP. In the final manuscript, we will more clearly state our goal of learning interpretable model outputs and features (in contrast to an interest in simpler/more interpretable architectures). We will also more clearly motivate our architecture choices.
>
> - What is the performance of MLP-NODE on the monkey Maze Dataset?
>
> Unsurprisingly, when trained on the Maze dataset MLP-NODE achieved similar reconstruction performance to ODIN at the matched dimensionalities (see Fig. R5). Unfortunately, at this time we can only quantify the benefits of ODIN over MLP-NODE (higher State $R^2$ when $\hat{D} > D$) when the underlying latent activity is observable.  We propose to qualitatively describe the MLP-NODE results in the main text and to explain that it is difficult to compare ODIN to MLP-NODE when underlying latent activity is unobservable.

---

> > ### Comment · Reviewer_dR2R · 2023-08-10
> >
> > I appreciate the extensive replies of the authors to myself and the other reviewers as well as the additional analyses they have performed.
> >
> > The addition of the 'invertible network' is a nice additional data point in favour of injectivity being a potentially useful inductive bias in these types of latent variable models. I am also glad to see that they quantified the 'invertibility' of the MLP model in their reply to reviewer UThX, and I would encourage reporting this in their updated manuscript.
> >
> > However, I have retained my original score for two primary reasons:
> > - while it is better to have three data points than two, this is still only circumstantial evidence that injectivity is important for latent variable models. If one were to e.g. compute a p value for "the two injective models perform better than the non-injective model", it would be ~0.25 under the null hypothesis that injectivity has no effect. It would be nice if it was e.g. possible to show that injectivity explains performance across or within model classes (are more invertible MLPs e.g. better than less invertible MLPs? and if some MLP instantiations are more invertible than some ODIN instantiations, do they perform better?).
> > - The only substantial performance difference demonstated between the MLP model and ODIN/'Invert' is in state R2 for toy synthetic problems, but the authors have not demonstrated how this translates to new insights for real neural data.

---

> > > ### Author Response · Authors · 2023-08-16
> > > **Response to comments from reviewer dR2R**
> > >
> > > We thank the reviewer for their prompt response, engagement, and insightful questions. We describe our analyses and results narratively in this response, but can also provide figures that illustrate these results if helpful. (Sharing figures might require coordination with the Area Chair.)
> > >
> > > - It would be nice if it was e.g. possible to show that injectivity explains performance across or within model classes (are more invertible MLPs e.g. better than less invertible MLPs?
> > >
> > > We agree, establishing a more direct link between injectivity and performance in latent variable models would be a powerful addition to the manuscript. To test for this link, we evaluated whether more invertible MLPs achieved higher State R2 performance by measuring cycle consistency on the $N=100$ MLP-NODE models depicted in Fig. R1 of our previous response. These models were all at $D=5$ latent dimensions and varied in the size and depth of the hidden layers of the readout MLP, thus achieving a wide range of State R2 ($0.77 \pm 0.13$). Indeed, differences in cycle consistency explained the majority of the variation in State R2 ($R^2 = 0.76$), indicating a very strong relationship between injectivity and performance of the MLP models. This result was also invariant to injecting noise when measuring cycle consistency.
> > >
> > > Including these results in a revised manuscript (e.g, in Fig. 2) would substantially strengthen the evidence that injectivity is a critical factor driving performance in latent variable models with nonlinear readouts.
> > >
> > >  - if some MLP instantiations are more invertible than some ODIN instantiations, do they perform better?).
> > >
> > > We agree such a relationship would provide additional evidence of the benefits of injectivity. However, since ODIN incorporates injectivity during optimization, it is very infrequent that ODIN models are less invertible than MLPs. For example, we measured cycle consistency on the ODIN models depicted in Fig. R1, and found only 1 out of the 100 models was less cycle consistent than the most cycle consistent MLP model. Similarly, because, ODIN models had a substantially higher state R2 performance with much lower variability than MLP-NODE ($0.92 \pm 0.02$ for ODIN, vs $0.77 \pm 0.13$ for MLP-NODE), it is hard to find enough MLP-NODE models with performance comparable to ODIN to make a meaningful comparison – likely owing to the difference in invertibility between the two models. Thus it would be challenging to do the suggested analysis in a meaningful way.
> > >
> > >
> > > - The only substantial performance difference demonstrated between the MLP model and ODIN/'Invert' is in state R2 for toy synthetic problems, but the authors have not demonstrated how this translates to new insights for real neural data.
> > >
> > > We are working to provide additional evidence on this point, and will post when available. We appreciate the reviewer’s engagement with the improvement of the manuscript.

---

> > > > ### Comment · Reviewer_dR2R · 2023-08-18
> > > >
> > > > I appreciate all the work the authors have done in response to the reviews. I agree that it has improved the paper and have updated my score accordingly.

---

### Official Review · Reviewer_GbAv · 2023-06-27

**Soundness:** 3 good
**Presentation:** 3 good
**Contribution:** 3 good
**Rating:** 7
**Confidence:** 4

**Summary:**

This manuscript looks at the challenge of capturing latent dynamics from neural data, a common modeling challenge in neuroscience that has attracted significant recent attention.   One common approach is to use deep learning methods, and recent highly predictive methods have been proposed that rely on high-dimensional latents to capture the structure.  This manuscript proposes to use an injective readout from the latent dynamics to encourage learning simpler and more realistic patterns from data, and qualitative results on the trajectories of the dynamics appear promising.

Updating to acknowledge the rebuttal. My review generally remains the same.

**Strengths:**

The major contribution of this manuscript is to focus on the readout layer of a commonly used model in neuroscience with promising results.  As this technique is largely generalizable, this is a useful trick that could be applied in many situations.

The experimental results look at many facets of the performance, including predictive performance on multiple tasks as well as the properties of the inferred dynamics and fixed points.  There is a worthwhile discussion on the greatest utility of machine learning methods in neuroscience, which often needs cleaner interpretation than slightly improved performance.  The fact that this method can get solid performance will a small latent dimension and capture real dynamics is promising.

**Weaknesses:**

The theoretical backing of this work is sparse, at best, and the proposed technique is very heuristic.  When will this technique be good enough?  When will it theoretically hold?  The long-term impact of this work would be greatly enhanced by answering these questions.

The experimental results are promising, but it is worth noting that there remains a gap in the predictive performance between a much more complicated model, AutoLFADS, and the proposed approach.  It would be very useful and expand on the contribution of the manuscript.

In the results, it appears AutoLFADS and GPFA are only shown with a single setting.  It would be worthwhile to examine how these models change as a function of their parameterization as well.

There should be a greater discussion of model selection, as it can be a difficult problem when considering many metrics of performance.

**Questions:**

Please show how the performance of AutoLFADs and GPFA vary as a function of their settings.

Please elaborate on the differences between AutoLFADs and ODIN, and explore the changes impact on performance.

**Limitations:**

The discussion seems largely fair, and there is no concern on negative societal impact.

Code is not included in the submission.  While the authors claim they will release if accepted, it would have been a stronger claim had anonymized code been included.

---

> ### Author Rebuttal · Authors · 2023-08-10
>
> Reviewer GbAv
>
> We thank the reviewer for their great suggestions to improve the manuscript.
>
> - The theoretical backing of this work is sparse, at best, and the proposed technique is very heuristic. When will this technique be good enough? When will it theoretically hold? The long-term impact of this work would be greatly enhanced by answering these questions.
>
> We appreciate the question and agree that theoretical guarantees will provide more exact conditions under which our results hold; unfortunately, we have not yet developed a theoretical framework with which to build guarantees. We hope that demonstrating the empirical benefits of injective readouts may inspire new theoretical efforts to understand to what extent and under what circumstances injectivity provides guarantees of latent system identifiability.
>
> - The experimental results are promising, but it is worth noting that there remains a gap in the predictive performance between a much more complicated model, AutoLFADS, and the proposed approach. It would be very useful and expand on the contribution of the manuscript.
> - In the results, it appears AutoLFADS and GPFA are only shown with a single setting. It would be worthwhile to examine how these models change as a function of their parameterization as well.
> - Please show how the performance of AutoLFADs and GPFA vary as a function of their settings.
> - Please elaborate on the differences between AutoLFADs and ODIN, and explore the changes impact on performance.
>
> We agree with these suggestions, and have trained additional AutoLFADS and GPFA models for direct comparison to ODIN at identical latent dimensionalities. In the AutoLFADS runs, we turned off input inference (intended for non-autonomous dynamics modeling, which is not considered in this manuscript), as the generator might offload part of the dynamics estimation to the controller RNN and cause the latents inferred by the generator to be misrepresentative. The remaining architectural differences between AutoLFADS and ODIN as tested here are RNN vs. NODE ($f$) and Linear-Exponential readout vs. Flow readout ($g$).
>
> While reconstruction performance can be an unreliable indicator of the accuracy of latent recovery, we know that models with poor reconstruction performance are likely missing out on important features of the neural data. Therefore, we view differences in reconstruction performance across model types as a flawed, but informative assessment of model quality (see Fig. R5). 5D ODIN models had dramatically higher co-bps (a measure of reconstruction performance) than 5D AutoLFADS models, and that 5D ODIN’s co-bps outperformed even 25D AutoLFADS models. Additionally, ODIN had almost 50% higher co-bps at the lowest dimensionality than GPFA had at the largest dimensionality. Of note, AutoLFADS performance reflects the best performing model across 20-worker evolutionary hyperparameter searches, which may provide AutoLFADS with an advantage over the single ODIN models. We propose to add these new results to Figure 5D (Fig. R5, excluding the MLP-NODE) and update section 4.4 to elaborate on the differences between ODIN and AutoLFADS (e.g., input inference, variational training, etc.).
>
> - There should be a greater discussion of model selection, as it can be a difficult problem when considering many metrics of performance.
>
> This is a good point. In our response to Reviewer UThX, we include a description of our procedure for hyperparameter searches conducted for this manuscript, as well as specific examples of hyperparameter sweeps that guided our model selection. These hyperparameter sweeps demonstrate a further advantage of ODIN over MLP-NODE, which is much greater robustness to hyperparameter choices. Additionally, we want to re-iterate two of the key results we showed for ODIN: robustness to choice of latent dimensionality, and maintaining correspondence between Spike NLL and State $R^2$. These improvements address critical model selection challenges with real biological data, where system dimensionality and State $R^2$ are unavailable. We propose to add two figures (Fig. R1, R2) to the supplement to demonstrate the effects of readout capacity and weight decay on ODIN and MLP-NODE. We will also clarify that ODIN enables Spike NLL and latent dimensionality to serve as model selection criteria.

---

### Official Review · Reviewer_x81c · 2023-07-05

**Soundness:** 2 fair
**Presentation:** 3 good
**Contribution:** 2 fair
**Rating:** 5
**Confidence:** 4

**Summary:**

The paper proposes Ordinary Differential equations autoencoder with Injective Nonlinear readout (ODIN) model, a model for inferring latent representations underlying high-dimensional neural spikings. Additionally, the ODIN model is designed to recover the latent dynamics and generative process. Through evaluations on synthetic and real neural datasets, ODIN is shown to be able to recover the true generative process.

**Strengths:**

- The paper is well written, with all details clearly stated for understanding the model and the experiments;
- The proposed model seems to work well on synthetic data;
- Visualisation of the fitted latents is sounding;

**Weaknesses:**

- The effect of injectiveness in the generative process is not evaluated;
- From Figure 2C, it seems like with three-dimensional latents, the linear model is able to yield similar performance as the full ODIN model (with normalising flow) in terms of latent inference, hence hindering the validity of the motivation of the paper, which is the non-linear embedding from the latents to the high-dimensional spikings;
- The proposed ODIN model is overly complicated comparing to standard neural manifold finding methods, with complicated neural architecture, I suspect related models would also yield similar high performance with comparable neural architecture and compute;

**Questions:**

See questions in Weaknesses.

**Limitations:**

Yes

---

> ### Author Rebuttal · Authors · 2023-08-10
>
> Reviewer x81c:
>
> We thank the reviewer for their suggestions. We believe that clarifying a few points will help to improve the reach and impact of the manuscript.
>
> - The effect of injectiveness in the generative process is not evaluated
>
> Unfortunately, we are not sure if this comment referred to the process by which the synthetic dataset was generated, or the injectivity of ODIN itself. If the former, a dataset in which the latent activity does not manifest fully in simulated neural activity would not match our understanding of the biological neural data, as it would imply features of the neural system which are not observable in the neural firing rates. If the latter, we test alternative injective readouts in our response to Reviewer dR2R and find that those readouts produce similar results to ODIN despite the only commonality between these architectures being injectivity.
>
> If we have misunderstood this point, we would be happy to respond in more detail with clarification from the reviewer.
>
> - From Figure 2C, it seems like with three-dimensional latents, the linear model is able to yield similar performance as the full ODIN model (with normalising flow) in terms of latent inference, hence hindering the validity of the motivation of the paper, which is the non-linear embedding from the latents to the high-dimensional spikings;
>
> We apologize for the confusion on this point. The State $R^2$ metric measures how well inferred latent states can be predicted from the true latent states. Thus, State $R^2$ can be high even when models do not capture the data well, such as if the inferred latent activity is a simple subspace of the true latent system. High State $R^2$ only implies good model performance when reconstruction performance (NLL and Rate $R^2$) is also good. In the case the reviewer mentions, the 3D Linear-NODE is not desirable because it did not achieve good reconstruction performance on the neural data. We will update section 3.4 to clarify that we require good performance with respect to both State R2 and Rate R2, as demonstrated by ODIN. See Fig. R4 for a qualitative summary.
>
> - The proposed ODIN model is overly complicated comparing to standard neural manifold finding methods, with complicated neural architecture, I suspect related models would also yield similar high performance with comparable neural architecture and compute;
>
> Unfortunately, it is difficult to address this concern in a way that would apply generally to the wide range of manifold estimation techniques in the field. We would be happy to test specific alternatives if the reviewer would like to see the result of a particular comparison. More generally, ODIN occupies a specific niche in the broader literature, belonging to a class of models that jointly learn latent dynamics $f$ and embedding $g$ to explain time-varying patterns of neural activity. In our paper, we test the common choices of $g$ (linear and MLP) and demonstrate that our approach leveraging injectivity offers substantial advantages over these previous methods. In this way, we believe the complexity introduced by the Flow readout is necessary for circumventing the issues with Linear and MLP readouts that we demonstrate in the manuscript.

---

### Official Review · Reviewer_UThX · 2023-07-06

**Soundness:** 3 good
**Presentation:** 4 excellent
**Contribution:** 2 fair
**Rating:** 5
**Confidence:** 4

**Summary:**

The paper presents ODIN, a new autoencoder model of population neural dynamics that uses a neural ODE (NODE) to model latent dynamics. The core innovation is the use of a simplified invertible ResNet to ensure that a nonlinear and approximately _injective_ readout from latent dynamics to population neural activity. The authors argue that approximate _injectivity_ encourages the model to learn simple and interpretable latent dynamics that can help explain the underlying neural computations. The authors evaluate their algorithm on both synthetic and real neural recordings, comparing its performance with related algorithms.

**Strengths:**

* The focus on readout injectivity is to the best of my knowledge novel in this field.
* The paper is well-written, with excellent and easy-to-understand figures.
* The authors evaluated their algorithms extensively, assessing the reconstruction of (i) neural activity, (ii) latents, (iii) neural activation function, and (iv) fixed-point reconstruction on synthetic and real neural recordings.

**Weaknesses:**

* It's unclear from the experiments whether ODIN significantly outperform MLP+NODE.
    * In figure 2C, the improvement of ODIN over MLP+NODE at the true dimensionality ($\hat{D}=3$) is small and it's unclear whether the improvement is significant. Some statistical tests here would make the argument more convincing. The claim that ODIN significantly outperforms MLP+NODE at $D>3$ is not convincing for me, as it would be fairly easy to pick the true dimensionality for MLP+NODE using State $R^2$ in Figure 2C.
    * MLP+ODE and ODIN both able to identify fixed-points accurately in Figure 3.
    * Figure 4C seems to be a clear case that shows ODIN outperform MLP+NODE. However, there are no error bars in the figure, which makes it hard to assess significance.
    * MLP-NODE is not evaluated in Figure 5?
* The paper could benefit from a more extensive discussion on how the hyperparameters were chosen. Did the authors conduct a hyperparameter search? My worry is that the small improvements of ODIN over MLP+NODE could be attributed to _unlucky_ hyperparameters.
* One common approach to encourage "simple" dynamics/readout is through weight regularization. The paper would be more convincing if it included more detailed discussions of/comparisons with such approaches.

**Questions:**

* In Figure S1, the authors showed the trained Neural ODEs are approximately injective. What would those same metrics be for Linear+NODE and MLP+NODE?

**Limitations:**

yes

---

> ### Author Rebuttal · Authors · 2023-08-10
>
> We thank the reviewer for their insightful feedback and their suggestions for improvements!
>
> -  the improvement of ODIN over MLP+NODE at the true dimensionality ($\hat{D}=3$) is small and it's unclear whether the improvement is significant.
>
> We apologize for the confusion: while the slight improvement in State $R^2$ of ODIN over MLP-NODE at $\hat{D}=3$ is a nice result (and statistically significant, t-test p-val < 0.01), the primary benefit of ODIN is that, unlike MLP-NODE, its latent accuracy is robust when $\hat{D} > D$.
>
> - it would be fairly easy to pick the true dimensionality for MLP+NODE using State $R^2$ in Figure 2C.
>
> Unfortunately, State $R^2$ is unavailable for real neural data, where “ground truth” latent activity is unobserved. Thus, it is critical that model performance be robust to incorrect choices of latent dimensionality, as is the case for ODIN but not MLP-NODE (Fig 2C). The reliability of ODIN’s latent recovery can provide confidence in the validity of the inferred latent activity when ODIN is applied to real neural recordings. *We propose* to clarify our explanation of this point in the manuscript and emphasize that spike negative log-likelihood will be the metric used for model selection on biological datasets.
>
> - MLP+ODE and ODIN both able to identify fixed-points accurately in Figure 3.
>
> The ability for MLP-NODE to find fixed points with similar accuracy to ODIN depends on the appropriate choice of $\hat{D}$ which cannot be determined for MLP-NODE.
>
> - Figure 4C seems to be a clear case... However, there are no error bars in the figure, which makes it hard to assess significance.
>
> Summary statistics would be beneficial here. We performed statistical tests and found that the activation functions inferred by ODIN had substantially higher $R^2$ than those inferred by either MLP-NODE or Linear-NODE for all neurons (N=60). These results were highly significant ($p<1e-10$ for both ODIN vs. MLP-,  Linear-NODE; paired t-tests). *We propose* to incorporate these statistics in a new panel to Figure 4.
>
> - MLP-NODE is not evaluated in Figure 5?
>
> MLP-NODE had similar Spike NLL to ODIN at matched state dimensionality (see Fig. R5). We did not include MLP-NODE on the Maze dataset because the benefits of ODIN over MLP-NODE are only visible when the underlying latent activity is observable. *We propose* to add a description of MLP-NODE’s performance on the Maze dataset in the text.
>
> - The paper could benefit from a more extensive discussion on how the hyperparameters were chosen.
>
> We came to our final hyperparameters via a series of searches over learning rate, generator size, encoder hidden size, and readout width and depth. To visualize one such sweep, we performed a new random search of 100 models with the readout hidden size drawn uniformly from [60,200] and layer depth uniformly from [1,3]. We set latent dimensionality to $\hat{D}=5$ (intentionally overestimated relative to the true system $D=3$) to assess the robustness of latent recovery.
>
> Results shown in Fig. R1. ODIN had higher and more stable State $R^2$ than MLP-NODE ($0.92 \pm 0.02$ vs. $0.77 \pm 0.13$). Spike NLL (the only observable metric for biological datasets) did not lend itself to choosing MLP-NODE HPs that produced optimal State $R^2$. In supplementary material, we will include a figure showing Spike NLL and State $R^2$ for a representative HP sweep.
>
> - One common approach to encourage "simple" dynamics/readout is through weight regularization.
>
> We had a similar idea, and have tested whether weight decay might encourage a simpler $\hat{f}$ and $\hat{g}$. We trained 100 MLP-NODEs with weight decay sampled from a log-uniform distribution from $[1e-8, 1e-4]$ (see Fig. R2). We found increasing the magnitude of weight decay to decrease State $R^2$. Spike NLL (red) was largely unaffected until it led to a breakdown in reconstruction performance. We propose an addition to the supplementary material that demonstrates the detrimental effects of weight decay on MLP-NODE latent recovery.
>
> - the authors showed the trained Neural ODEs are approximately injective. What would those same metrics be for Linear+NODE and MLP+NODE?
>
> This is an interesting question. To test whether the MLP was also approximately injective, we estimated the invertibility of the learned mapping by measuring cycle-consistency (see Fig. R3). Cycle-consistency quantifies how accurately the input to a function can be recovered from its output; in this case, how well the inferred latent activity can be recovered from predicted firing rates. To estimate this reverse mapping, we trained a separate MLP to predict the latent activity corresponding to the inferred log-rates for each model. The latent activity of a 10D ODIN model was recovered with higher accuracy than for a 10D MLP-NODE ($R^2$ of 0.995, 0.912 respectively). This suggests that ODIN is substantially closer to being truly injective than MLP-NODE.
>
> We wondered whether MLP-NODE might also learn to compress features of latent activity so that these components have negligible effects on predicted firing rates. We added Gaussian noise to the inferred rates and measured the effect of noise on cycle-consistency. The more a readout compresses latent activity, the greater the instability in its reverse mapping. In practice, MLP-NODE was far less cycle-consistent than ODIN at all levels of noise. ODIN seems to learn a more nearly injective mapping from latent activity to firing rates than MLP-NODE.
>
> We show in our response to Reviewer dR2R that Linear-NODE models with more than 3 state dimensions learn non-injective mappings from latent to neural space when the 3D data is linearly-embedded. In contrast with MLP- and Linear-NODE, ODIN’s approximate injectivity forces superfluous latent activity to damage reconstruction performance. We propose to add these injectivity quantifications to the supplementary material to give readers more information on the relative injectivity of the different readouts.

---

> > ### Comment · Reviewer_UThX · 2023-08-21
> >
> > I thank the authors for their extensive response. They have addressed many of the initial concerns that I have and I have thus adjusted my scores accordingly.
> >
> > I still have mild concerns over the utility of this approach given that there might be other ways to identify the _optimal_ dimensionality. For example, in Figure 2, judging purely from R^2 and Spike NLL, one might still be inclined to choose $\hat{D}=3$ as the dimensionality for MLP-NODE as it represents the "elbow". Furthermore, if you have an additional variable (e.g., hand kinematics) to regress against, one might be able to choose the optimal dimensionality that way.
> >
> > I'm not familiar with Cycle-consistency and so I don't have too much to add on the subject. However, it would be useful if the authors could also quantify significance  over the reconstructed latent R^2, as well as how this might depend on the expressiveness of the separate MLP that the authors trained to predict the inferred latent activity. Would for example a bigger network and/or more data close the gap between MLP-ODE and ODIN? And if so, how should we think about cycle-consistency as a measure of invertibility?

---

### Author Rebuttal · Authors · 2023-08-10

*General Response:*

We thank the reviewers for their attention and feedback, which has allowed us to strengthen this body of work.

Overall, reviewers pointed out that our focus on readout injectivity was a novel approach to an important problem in neural data modeling (UThX, GbAv). They pointed out that our submission was well-written and detailed (UThX, x81c) with easy-to-understand figures (UThX, x81c) and exhaustive experimental evaluations (UThX, GbAv, dR2R). They agreed on the model’s solid performance (x81c, GbAv, dR2R) and pointed out that our approach is useful and generalizable (GbAv).

There were a few common categories of criticism that we would like to address up front.

- *Benefits over existing methods:*

Some reviewers raised questions about the advantages of ODIN over alternatives Linear-NODE (x81c) and MLP-NODE (UThX) and over existing methods AutoLFADS and GPFA (GbAv). In addressing these criticisms, we clarified that ODIN’s advantage over Linear-NODE is that it achieves better performance in both State $R^2$ and NLL, and its advantage over MLP-NODE is that it is more robust to overestimates of the latent dimensionality $D$. As $D$ is often unknown in real datasets, this robustness provides additional confidence in the validity of inferred latent activity. Finally, we demonstrate on a biological dataset that ODIN has substantially better performance at low dimensionality than either AutoLFADS or GPFA.

- *Simpler modifications to existing methods:*

Reviewers were also interested in our choice of architecture (x81c, dR2R) and whether a simpler approach like weight decay or other hyperparameter tuning could help mitigate issues with existing models (UThX, GbAv). In response, we outlined the class of models to which ODIN belongs, enumerated alternatives, and explained why they do not meet our modeling needs. We performed a random search to demonstrate the robustness of State $R^2$ for ODIN and its fragility for the MLP-NODE. Finally, we trained MLP-NODE with a wide range of weight decay settings and showed that this did not improve State $R^2$.

- *Theoretical basis:*

One reviewer requested theoretical guarantees for our approach (GbAv), and others pointed out that injectivity is possible for existing methods (UThX) and questioned whether the potential lack of injectivity was the real problem (dR2R). We acknowledge that this experimental work does not address the theory behind the success of ODIN, and we hope to explore that in future work. We supported our claim that injectivity was the source of ODIN’s beneficial properties by replicating our results with an alternative injective readout. We added an analysis of effective rank to illustrate problems with linear readouts that may be technically injective. Finally, we performed an additional experiment comparing the cycle consistency of ODIN and MLP-NODE in the presence of noise.

We are excited about the new analyses that came about thanks to reviewer criticisms and look forward to engaging with any additional comments.

---

### Decision · Program_Chairs · 2023-09-21

**Decision:**

Reject

**Comment:**

This work uses an MLP model of dynamics within an autoencoder to learn representations of dynamical systems. A number of tricks in training are used to stabilize the system, and a number of examples demonstrate how the proposed model can capture features of the data, such as fixed points.

While reviewer GbAv gave a more favorable scores, most of the reviewers retained a borderline stance on the paper due to combination of the complexity of the model as compared to the potential that the main issues aimed at being solved (e.g., injectivity) are not actually the bottleneck in the systems. The reviewers are thus convinced that additional motivation and demonstration that the method is truly solving an important problem in dynamics learning is necessary. I thus do not recommend this work for acceptance.